# SoftBiT: Soft Bimanual Teleoperation with Proprioceptive Visual Augmentation

Tai Inui
*Waseda University*
Tokyo, Japan
taiinui556@suou.waseda.jp

Uksang Yoo
*Carnegie Mellon University*
Pittsburgh, PA, USA
uksang@cmu.edu

Jeffrey Ichnowski
*Carnegie Mellon University*
Pittsburgh, PA, USA
jeffi@cmu.edu

Jean Oh
*Carnegie Mellon University*
Pittsburgh, PA, USA
jeanoh@cmu.edu

*Abstract*—Soft robotic teleoperation offers unique advantages for bimanual manipulation, but users often struggle with visual feedback during operation, particularly when robot fingers are occluded by objects. We introduce SoftBiT, a teleoperation interface that enhances user awareness through real-time soft robot finger shape visualization in extended reality (XR). Our system combines proprioceptive sensing with an XR headset (Meta Quest 2) to provide users with intuitive visual feedback about finger deformations during manipulation tasks. SoftBiT's key innovation is a real-time sim-to-real pipeline that estimates and visualizes soft finger shapes, helping users better understand robot-object interactions even when direct visual feedback is limited. Through three representative tasks (pick-and-place, assembly, and object deformation), we demonstrate how augmented proprioceptive feedback supports user decision-making during manipulation. Our shape estimation system achieves 42.55 FPS, enabling smooth real-time visualization. This work lays the foundation for future user studies investigating how proprioceptive augmentation impacts teleoperation performance and user experience, with potential extensions to more complex multi-fingered manipulation tasks.

*Index Terms*—Human-Robot Interaction, Teleoperation, Proprioception, Virtual Reality, Soft Robotics

## I. INTRODUCTION

Effective teleoperation of robotic systems relies heavily on clear and immediate feedback between the user and robot. This feedback becomes particularly crucial when manipulating objects with bimanual robot systems, where direct observation of robot-object interactions is often limited by self-occlusion. A fundamental challenge in teleoperation is intelligibility - enabling users to accurately interpret the robot's state and behavior in real-time [1]. Without adequate feedback, users struggle to predict and control robot movements, leading to reduced manipulation accuracy and efficiency.

Research in teleoperation interfaces has explored various approaches to improve user feedback, from conventional haptic interfaces to immersive technologies [2], [3]. Extended reality (XR) interfaces, in particular, have shown promise in enhancing spatial understanding and control precision during manipulation tasks. Concurrent advances in soft robotics have introduced new opportunities for user feedback: soft robots not only provide passive compliance for safer interaction, but their deformable structure naturally visualizes contact forces and object interactions [4]. However, no system has

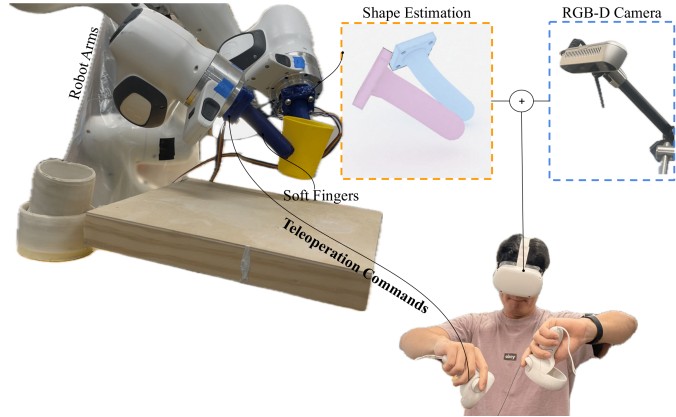

Fig. 1. SoftBiT: An intuitive XR bimanual robot teleoperation system integrating soft end-effector shape estimation into the user's view.

integrated these deformation cues into immersive interfaces for intuitive teleoperation.

This workshop paper presents SoftBiT, a soft bimanual teleoperation interface that enhances user awareness by visualizing finger deformations in extended reality. Our key insight is that soft robot fingers naturally encode interaction information through their shape changes, providing an intuitive visual signal of contact conditions and applied forces. SoftBiT leverages this property by combining proprioceptive sensing with an XR headset (Meta Quest 2) to overlay real-time finger shape estimations onto the user's view. Our sim-to-real pipeline processes data from embedded strain sensors to provide responsive visual updates that support fluid interaction, even during occluded manipulation.

We evaluate the system through three representative scenarios: pick-and-place, assembly, and deformable object manipulation, demonstrating how this enhanced visual feedback supports user understanding and control. By combining proprioceptive sensing with intuitive XR visualization, SoftBiT aims to make robot teleoperation more transparent and predictable for users. Our immediate future work will investigate how this augmented proprioceptive feedback influences demonstration quality through user studies and trained imitation learning policy performance.

## II. RELATED WORKS

Interest in teleoperation for robot manipulation has grown rapidly in recent years [5], driven by two key factors. First, the increasing demand for remote work in industrial settings has spurred development of more intuitive teleoperation interfaces [6], [7]. Second, the rise of imitation learning approaches for training autonomous robot policies, such as diffusion policies [8]–[10], requires high-quality human demonstrations. Both applications emphasize the need for interfaces that support extended periods of precise manipulation while maintaining user comfort and situational awareness.

Research in robot teleoperation interfaces has explored various approaches to enhance user feedback and control. Conventional solutions have focused on haptic interfaces [11], while recent works increasingly leverage immersive technologies like virtual and augmented reality to improve spatial understanding [12].

These developments in interface design complement recent advances in soft robotics. Soft robots offer inherent advantages for user interaction: their compliance provides passive adaptation during manipulation and natural error tolerance [13], while their deformable structure visually encodes interaction forces and contact conditions [4]. When combined with proprioceptive sensing, this deformation behavior provides a rich source of feedback that could help users understand robot-object interactions.

While prior work has demonstrated the benefits of both immersive interfaces and soft robotics independently, the potential of combining these approaches - particularly using soft robot deformation as an intuitive visual feedback channel in XR - remains largely unexplored. Our work bridges this gap by integrating proprioceptive sensing and XR visualization to enhance user awareness during soft robot teleoperation.

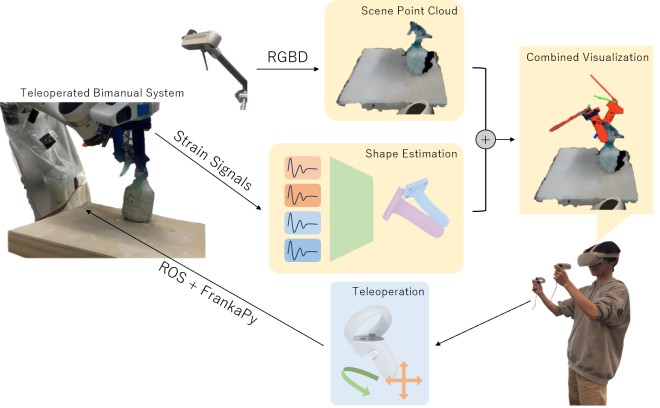

Fig. 2. System Overview: SoftBiT creates an intuitive XR robot teleoperation environment by integrating the estimated shape of the soft end-effectors into the scene.

## III. METHOD

### A. Hardware

The SoftBiT system is composed of two 7-axis robot arms (Franka Emika Panda Robot) with soft end-effectors, each embedded with four strain sensors. Both robot arms are teleoperated via an XR headset (Meta Quest 2), allowing for immersive and intuitive manipulation. Two RGB-D cameras capture the interaction scene, providing the base context for visual augmentation via proprioception.

### B. Teleoperation

In SoftBiT, two Franka Panda robot arms are controlled simultaneously in real-time using inputs from XR controllers. Initially, the complete pose (translation and rotation) of each XR controller is transmitted from the headset to a PC via TCP. A PID controller then minimizes the error between the received controller pose and an internally updated reference pose. At each control loop, the PID output is applied directly to the robot's end-effector's target pose. The target pose is then converted to target joint angles with inverse kinematics (IK). Finally, the target joint angles are executed on the robot arms in real-time using the FrankaPy [14] controller. By ensuring steady, low-latency teleoperation, this processing pipeline makes it possible to guide each arm intuitively using the appropriate XR controller.

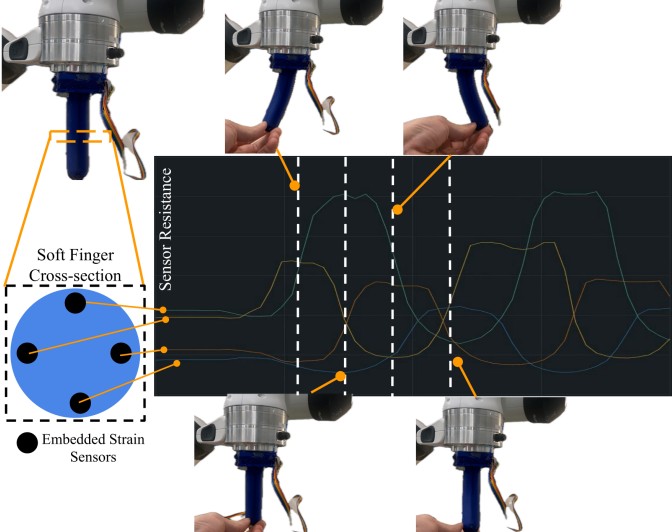

Fig. 3. Soft robot end-effector with proprioceptive sensors

### C. Proprioceptive Sensing and Calibration

As illustrated in Figure 3, there are four strain sensors embedded in each of the soft fingers, which record the change of resistance caused by the deformation of the fingers. We implement a sim-to-real pipeline based on finite element method (FEM) simulations and a proprioceptive network to infer the corresponding shape of the fingers from these sensor readings. This pipeline is shown in Figure 4, taking the following steps:

- **Simulated Training Data Generation:** We create a synthetic dataset of soft finger deformations with their corresponding strain sensor readings using a FEM-based simulator.
- **Proprioceptive Model Training:** The simulated dataset is used to train a proprioceptive network [15] which learns to map strain sensor readings to estimations of soft finger shape.
- **Sensor Calibration:** Covariance Matrix Adaptation Evolution Strategy (CMA-ES) is used to adjust sensor biases and scale factors to ensure consistency between the simulation and real-world sensor signals.
- **Real-Time Inference:** After calibration and training, the model can be applied for live inference with 42.55 FPS, predicting the deformation of the finger based on the sensor input.

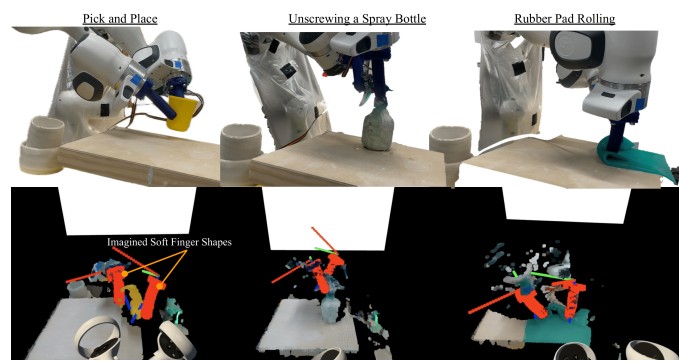

Fig. 5. Teleoperation evaluation tasks: pick-and-place (left), assembly (middle), and deformable object manipulation (right).

## IV. MEASURES AND ANALYSIS

To understand how proprioceptive feedback enhances teleoperation, we evaluated SoftBiT across three manipulation scenarios that present distinct visual feedback challenges:

- **Pick-and-Place:** Users performed grasping tasks where objects typically occlude the contact points. The shape estimation overlay provided continuous feedback about grip state despite visual obstruction, helping operators maintain stable grasps.
- **Assembly (Unscrewing a Spray Bottle):** This task required precise control and sustained contact. Real-time visualization of finger deformation helped users maintain appropriate force levels throughout the unscrewing motion, especially when the contact point moved out of direct view.
- **Deforming Objects (Rolling a Rubber Pad):** Users manipulated a compliant rubber pad, where successful execution required understanding both the robot's grip state and the object's deformation. Shape feedback provided crucial information about the interaction forces and material response.

Initial testing suggests that augmented proprioceptive feedback enhances users' understanding of robot-object interactions. To quantitatively evaluate this system's impact, we propose a future user study examining:

- **Usability:** How does proprioceptive augmentation influence objective performance metrics (task completion time, success rate) and subjective measures (user confidence, perceived control, comfort with the system)?
- **Feedback Integration:** What visualization strategies most effectively complement natural visual feedback while minimizing cognitive load? How do different levels of augmentation affect user attention and task awareness?
- **Demonstration Quality:** How does enhanced visual feedback affect the quality of recorded demonstrations for imitation learning, as measured by the performance of policies trained on user demonstrations?

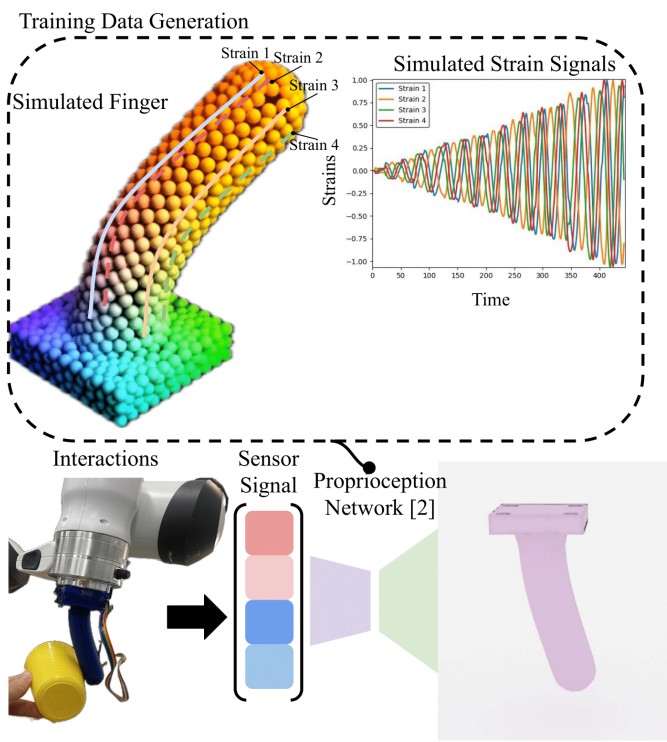

Fig. 4. Sim-to-real soft finger proprioception pipeline.

### D. Proprioceptive-Augmented Visual Feedback

To mitigate visual occlusion and improve user awareness of robot-object interactions, SoftBiT overlays real-time soft finger shape estimations onto the XR interface. The 3D shape of each finger is estimated in real-time based on strain sensor readings, then rendered as a point cloud overlay in the XR headset. As a result, this allows users to see the soft end-effector shape without occlusion, along with the spatial relationship between the end-effectors and the interactive scene obtained from the RGB-D cameras.

## V. Conclusion

This paper presents SoftBiT, a teleoperation interface that enhances user awareness during bimanual manipulation through proprioceptive-augmented visual feedback. By combining soft robotic end-effectors with real-time shape estimation and XR visualization, our system helps users better understand and control robot-object interactions, particularly during visually occluded manipulation tasks. The system achieves responsive 42.55 FPS shape estimation, enabling fluid interaction during complex manipulation scenarios such as pick-and-place, assembly, and deformable object manipulation.

Our initial demonstrations suggest that proprioceptive augmentation can meaningfully enhance teleoperation by providing users with continuous feedback about robot-object interactions, even when direct visual feedback is limited. This work opens several promising directions for future research. We plan to conduct user studies to quantify how proprioceptive augmentation affects task performance, user confidence, and cognitive load. Additionally, we will investigate how enhanced visual feedback influences the quality of demonstrations for imitation learning, and explore extensions to more complex manipulation scenarios.

## Acknowledgment

This work is supported by the NSF GRFP (Grant No. DGE2140739) and MOTIE, Korea (Grant No. 20018112).

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
