# OpenReview forum: "SoftBiT: Soft Bimanual Teleoperation with Proprioceptive Visual Augmentation"
_humanrobotinteraction.org/HRI/2025/Workshop/VAM — HRI 2025 Workshop VAM Submission_

### Official Review · Reviewer_zSYL · 2025-02-28

**Rating:** 7
**Confidence:** 5

**Review:**

This is a good paper that is highly relevant to the workshop. I look forward to hearing more about this project.

However, here are a couple of suggestions to improve the paper and its readability
- Insert figures as they are mentioned in the text, it is inconvenient for the reader that figures are inserted a page before they are discussed
- The significance and need for this contribution could be better justified
- If possible, expand the description of the analysis. Currently, this is done superficially and gives only a vague impression of how the work was evaluated and tested

---

### Decision · Program_Chairs · 2025-02-26

Accept